# Misam: Using ML in Dataflow Selection of Sparse-Sparse Matrix Multiplication

Sanjali Yadav*, Bahar Asgari*
*University of Maryland
*sanjali7@umd.edu, bahar@umd.edu*

*Abstract*—Sparse matrix-matrix multiplication (SpGEMM) is a critical operation in numerous fields, including scientific computing, graph analytics, and deep learning. These applications exploit the sparsity of matrices to reduce storage and computational demands. However, the irregular structure of sparse matrices poses significant challenges for performance optimization. Traditional hardware accelerators are tailored for specific sparsity patterns with fixed dataflow schemes—inner, outer, and row-wise—but often perform suboptimally when the actual sparsity deviates from these predetermined patterns. As the use of SpGEMM expands across various domains, each with distinct sparsity characteristics, the demand for hardware accelerators that can efficiently handle a range of sparsity patterns is increasing. This paper presents a machine learning-based approach for adaptively selecting the most appropriate dataflow scheme for SpGEMM tasks with diverse sparsity patterns. By employing decision trees and deep reinforcement learning, we explore the potential of these techniques to surpass heuristic-based methods in identifying optimal dataflow schemes. We evaluate our models by comparing their performance with that of a heuristic, highlighting the strengths and weaknesses of each approach. Our findings suggest that using machine learning for dynamic dataflow selection in hardware accelerators can provide upto 28× gains.

## I. INTRODUCTION

This document provides instructions for submitting papers to ISCA 2023. In an effort to respect the efforts of reviewers and in the interest of fairness to all prospective authors, we request that all submissions to ISCA 2023 follow the formatting and submission rules detailed below. Submissions that violate these instructions may not be reviewed, at the discretion of the program chair, in order to maintain a review process that is fair to all potential authors. This document is itself formatted using the ISCA 2023 submission format. The content of this document mirrors that of the submission instructions that appear on the conference website. All questions regarding paper formatting and submission should be directed to the program chair.

## II. INTRODUCTION

Sparse matrix-matrix multiplication (SpGEMM) is a key operation in many scientific computing, graph analytic and deep learning applications. While sparse matrices reduce storage and operation costs, their inherent irregular structure poses a challenge towards achieving high performance processing due to low utilization of both computation resource and memory bandwidth. Prior work addressed these challenges by developing various hardware accelerators customized for the three

widely recognized SpGEMM execution dataflow schemes: inner product (IP) [1], [5], outer product (OP) [4], [8], and row-wise product (RW) [6], [7] . However, these accelerators are optimized for specific sparsity patterns that best utilize their underlying hardware architectures. They employ a fixed execution dataflow, which optimizes the input or output data reuse, at the expense of the other. As a result, the performance is sub-optimal if the sparsity of the workload does not align with the rigid design of the accelerator.

As SpGEMM becomes more prevalent in an array of application domains, the demand for hardware accelerators capable of effectively handling a broad spectrum of sparse patterns is on the rise. For example, graph datasets are usually extremely sparse with only 0.001% nonzeros whereas neural network weight matrices are usually much smaller but relatively denser. The current approaches usually develop a dedicated hardware accelerator with a predetermined execution dataflow optimized for a particular domain. However, to achieve a more universal hardware, we require a mechanism to select the best dataflow for diverse workloads across different domains.

Recent studies such as Spada [2] and Flexagon [3] recognize the tradeoff of having a fixed execution dataflow design and propose hardware accelerators that can dynamically adapt the dataflow scheme to support workloads with varying sparsity patterns. Spada proposed a window-based adaption algorithm that divides the rows of input matrix A into bands and profiles the performance results on the first few rows of the band to determine the optimal dataflow. During the profiling phase, different window shapes are tested on the band and if there is a decrease in performance, their method stops profiling and selects best window found at that point. This is a limitation of this approach as it may lead to sub-optimal selection incase of insufficient profiling. Similarly, Flexagon implements a simple profiling method that uses features of the SpGEMM operation to be executed (i.e., matrix dimensions and sparsity patterns) and determines the best dataflow. They defer the identification of the optimal method for analyzing data flow to future research.

The current state-of-the-art dataflow selection process comprises of selecting features that can represent each scheme and developing a heuristic that uses these features to guides an optimal selection process. However, heuristic maybe not provide the most optimal solution as seen in Spada's case and it often suffer from lack of generalization. Interestingly, the character-

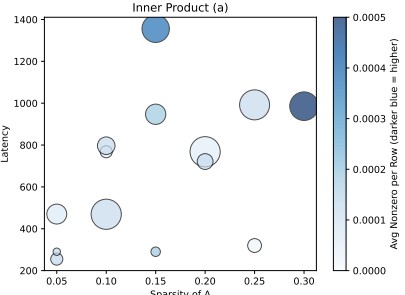 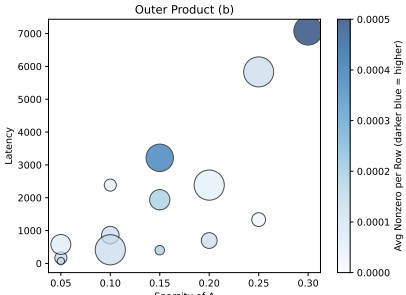 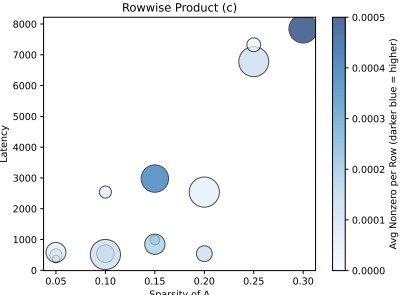

Fig. 1: **Sparsity Analysis –** Illustrates the relationship between sparsity of input matrix A (x-axis), sparsity of input matrix B (size of the bubble), average number of nonzero per-row in matrix A (color depth) and the latency (total number of cycles) of the three dataflow schemes for various SpGEMM experiments.

istics of this problem are well-suited to machine learning (ML) techniques commonly employed in data classification – *given the features of the input matrices, we can categorize them into classes corresponding to different dataflow schemes*. Our work, Misam (/'maIz@m/), is motivated to explore whether employing ML-based mechanism could offer a more viable and optimal alternative, and we aim to assess any potential trade-offs associated with this approach.

## III. MOTIVATION

Features or inputs play a pivotal role in the performance of ML models. However, there is no established literature guiding the selection of optimal features for addressing this type of problem. Therefore, we conducted a preliminary analysis to explore how certain well-known features of input matrices affect the latency of each dataflow scheme as illustrated in Figure 1. We focused on the sparsity of input matrices A and B and the average number of non-zero entries per row in matrix A, as these are the most prominent features in profiling methods. Our findings indicate that as both matrices become denser, the inner product generally outperforms the other schemes, since the outer and rowwise products must handle the summation of substantial partial results. In contrast, when the matrices are sparser, the outer and rowwise products often perform as well or better. Additionally, an increase in the average number of non-zero entries per row tends to degrade the performance of the rowwise product, likely due to inefficiencies in fetching corresponding rows of matrix B from the main memory. However, we noticed an absence of strong correlations in the three graphs likely suggesting the presence of confounding variables. This motivated us to further investigate into which features most significantly dictate the relationship between the optimal dataflow scheme and the input matrices and incorporate those in our ML model.

Regarding the type of ML model, we focus on decision trees and deep reinforcement learning. Decision trees are chosen for their lightweight nature and widespread use in classification tasks, offering an efficient and straightforward approach. On the other hand, deep reinforcement learning, despite its higher computational overhead, is selected for its ability to more ac-

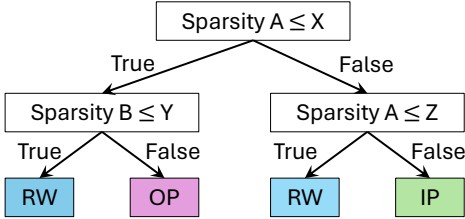

Fig. 2: **Decision Tree Structure –** Decision-making involves navigating from the root to the relevant leaf, assessing conditions at each node along the path.

curately capture the complexities of the problem. Our objective is to identify the most suitable features for our models and also utilize them to construct a heuristic. Subsequently, we assess the strengths and weaknesses associated with employing a decision tree model, reinforcement learning model, and simple heuristic, and present our conclusions.

## IV. BACKGROUND

*Decision Trees–* A fundamental tool in predictive modeling uses the values of input features to partition the feature space into regions for making predictions. A decision tree is a supervised learning algorithm characterized by a hierarchical tree structure, which models decisions and their potential outcomes. The process begins at the root node, which encompasses the entire dataset. Subsequent internal nodes, or decision nodes, correspond to specific features within the dataset. These nodes facilitate the partitioning of data into two or more subsets based on specific conditions. The branches of the tree signify the outcomes of these conditions, leading to additional sub-nodes. The process culminates at the leaf nodes, each representing the final outcome of the decision process as illustrated in Figure 2.

*Reinforcement Learning–* Reinforcement Learning (RL) is a type of unsupervised machine learning algorithm where an agent interacts with an environment by taking actions at each time step to earn rewards as illustrated in Figure 3. In decision trees, each data point in the dataset must be labeled with the correct outcome, which allows us to supervise the model to

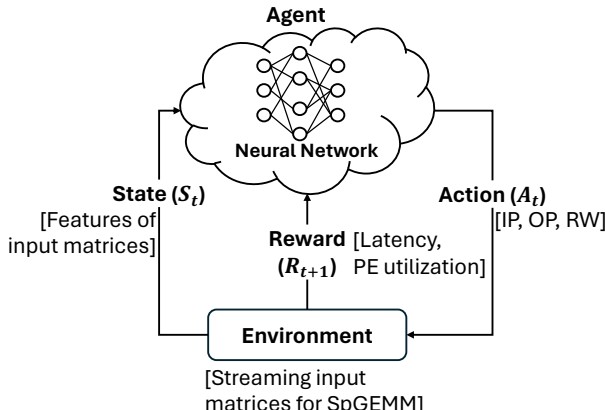

Fig. 3: **RL Structure –** At each timestep, the agent perceives the current state and uses its knowledge base to take an action. A reward is assigned to provide feedback.

select the optimal classification. Conversely, in RL, the agent is not explicitly instructed on the best outcome; instead, it is guided by a reward function. The objective of the agent is to establish an optimal policy that associates environmental states with actions that the agent should undertake to maximize the expected cumulative reward, commonly referred to as the Q-value. The Q-value, denoted by Q(s, a), represents the expected reward for selecting a specific action in a particular state, reflecting the agent's current knowledge of the environment. Traditional RL utilizes a Q-table to record and update these Q-values for each state-action pair. Initially, Q-values are randomly assigned but are refined through a process known as Q-learning, where values converge as the agent learns from ongoing interactions.

Given the complexity of our problem and the high-dimensional nature of our input data, a simple Q-table is insufficient. Instead, we employ a Deep Q-Network (DQN), which leverages deep neural networks to approximate the Q-value function. Training the network involves continuously adjusting the Q-values based on the reward feedback and a temporal difference error method, which assesses the disparity between predicted and observed Q-values after selecting an action. We further enhance the learning stability by employing experience replay, where the network learns from a diverse range of past experiences and breaks the temporal correlations between consecutive learning samples, thereby improving learning efficacy and policy performance.

## V. METHODOLOGY

*Dataset Creation–* To train and evaluate our machine learning models we require a dataset. Unfortunately, there are no existing datasets containing sparse matrices along with information about various features of the matrix. Thus, we create our own dataset with 30 matrices from SuiteSparse belonging to various domains such as graph analytics, machine learning, linear programming, chemical process simulation, etc with different dimensions and sparsity patterns. We also randomly generate 70 matrices with random patterns and varying

sparsity. Some of these benchmarks are described in Table I. We have developed three different cycle accurate simulators for inner product, outer product and row-wise product with a fixed PE size of four. The input matrices A and B are divided into smaller blocks and streamed into memory, partitioned in a way to enable multiplication without dependencies on other blocks. For computations, matrix A is formatted in CSR and matrix B in CSC for the IP, while for the OP, their formats are reversed. For RW, both matrices are maintained in CSR format. The matrices belonging to the same domain (i.e., graph analytics, neural-network) are either multiplied with similar matrices of matching dimensions or with themselves via the three different execution dataflow schemes. In cases where the dimensions of the input matrices did not match, the larger matrix was trimmed to fit the size of the smaller one. The latency of this multiplication process and the PE utilization (defined as the average number of busy cycles for the PEs) are recorded in the dataset along with various features of the input matrices mentioned in Table II.

*Feature Selection–* All features, except for blocks_accessed, can be directly derived from the sparse storage formats such as CSR and CSC, requiring no preliminary processing. The blocks_accessed feature tracks the number of blocks needed for the rowwise product that are not currently stored in memory. To calculate this, we analyze the distribution of columns in input matrix A, which is accessible in CSR format and estimate the number of columns distant from the current column to determine the blocks of rows from matrix B that need to be loaded into memory. Out of the 12 features in Table II, we must select those of greatest significance for our model because including all features would increase the model's inference latency and storage size. Therefore, we employ the 'sklearn' library in Python to construct a decision tree incorporating all features. This approach allows us to identify and analyze the most influential features for classification, as demonstrated in Figure 4.

*Decision Trees–* We select the top five features for our

TABLE I: **Benchmarks**

| ID | Matrix | Sparsity | ID | Matrix | Sparsity |
|----|--------|----------|----|--------|----------|
| en | email-Enron | 2.73e-4 | rg | rgg_n | 1.25e-5 |
| wb | webbase-1M | 3.1e-6 | cs | 2cubes_sphere | 1.6e-4 |
| bs | boneS01 | 3.4e-4 | cg | cage12 | 1.2e-4 |
| s1 | sherman1 | 3e-3 | o1 | olm1000 | 3e-3 |
| pp | p2pGnutella31 | 3e-5 | tl | t3dl | 1e-3 |
| bc | bccstm39 | 2e-5 | cc | cca | 5e-5 |
| bu | bauru5727 | 8e-5 | wg | web-Google | 6e-4 |
| wb | webbase-1M | 3e-6 | mn | mnist_test_norm | 1e-3 |
| t20 | test20 | 1e-3 | t40 | test40 | 2e-3 |
| t30 | test30 | 9e-4 | t57 | test57 | 4e-4 |

TABLE II: **Features**

| Feature Name | Description |
|--------------|-------------|
| sparsity | Total nonzero in matrix /size of matrix |
| avg_row_length | Average number of nonzero per row |
| avg_col_length | Average number of nonzero per column |
| avg_row_length_var | Variance in average number of nonzero per row |
| avg_col_length_var | Variance in average number of nonzero per column |
| blocks_accessed | Blocks of rows accessed not in memory |
| size | Size of matrix block |

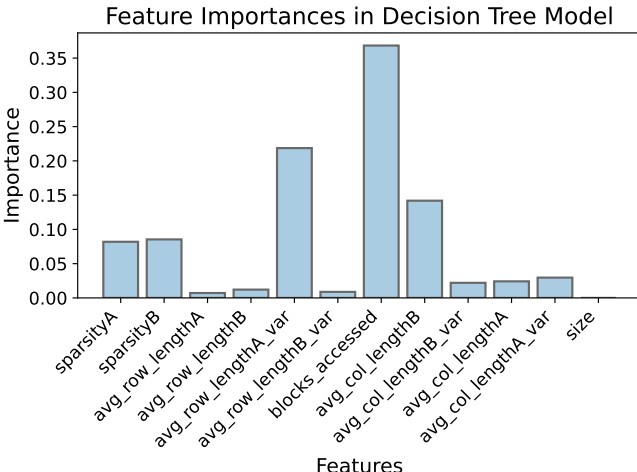

Fig. 4: **Feature Selection** – Analysis of the feature importance in the decision tree.

decision tree model as identified in Figure 4: blocks_accessed, avg_col_lengthB, avg_row_lengthA_var, sparsityA, and sparsityB. Our dataset is partitioned into a training set comprising 70% of the data and a validation set comprising the remaining 30%. Each matrix within these sets is segmented into smaller blocks for streaming purposes, with each block represented as a row in our dataset. As decision trees operate under supervised learning, we labeled each row, corresponding to an SpGEMM operation between matrix blocks, with the optimal dataflow algorithm determined by latency measures. We also observe an imbalance of class representation in the training dataset, which could introduce bias in the model towards majority class. We use a weight balancing technique, where each class is assigned a weight that is inversely proportional to its frequency in the training data to ensure better detection rates for the minority class and a more balanced model.

The decision tree model was created using 'sklearn' library in python. To reduce the risk of the model overfitting, a common issue with decision trees, we implemented a pruning condition that restricts the tree depth to 9. This depth was chosen after testing various depths, as it offered a good compromise between minimizing the decision tree's storage size and maximizing its accuracy. Furthermore, we used k-fold cross validation technique to evaluate the model's performance and its robustness.

*Reinforcement Learning*– The RL model, as depicted in Figure 3, includes three core components: state, action, and reward, operating within a defined environment. Our environment is set at the hardware level where an agent predicts the optimal dataflow scheme for processing streaming input matrices. The state is defined by the feature set used in our decision tree analysis, while the potential actions involve executing SPGEMM through inner, outer, or rowwise products. Upon receiving input matrices, the agent analyzes the relevant features to determine the most efficient dataflow algorithm. Rewards are given based on the agent's ability

to choose the scheme that achieves the lowest latency and the best PE utilization, with these performance metrics drawn from our dataset that records latencies for all three methods. Currently, our model is trained offline to enable comparison with the decision tree, intended solely for inference without a active learning component if deployed. To facilitate an online learning strategy, we need to modify the reward function to enable dynamic assignment during runtime as new data arrives. This adaptation will be the focus of our future research efforts.

The core of our model is a neural network designed to determine the appropriate action to take upon encountering a specific set of features, or the current state. To optimize storage, our neural network features just one hidden layer and contains 9,219 parameters. We utilize an epsilon-greedy policy, initially setting epsilon to a high value to encourage the agent to explore a wide array of actions, thereby avoiding local optima. As training progresses, we gradually reduce the epsilon value through a decay function, allowing the agent to increasingly rely on its accumulated knowledge (exploitation) over random choices for decision-making. To assess the model's effectiveness, we adopt an approach similar to that used with decision trees: dividing the dataset into training and evaluation sets. The agent learns from the training set and is subsequently evaluated based on its performance with unseen feature sets in the evaluation set.

*Decision-Tree-Guided Heuristic*– We also evaluate our models against a simple heuristic. In the absence of established state-of-the-art heuristics, we have created one based on our decision tree, limiting its depth to two levels and transforming it into nested if-else statements as shown in Listing 1. The most critical features are positioned at the top of the decision tree because they play a pivotal role in decision-making. Consequently, our heuristic, which is essentially a condensed version of the original decision tree, focuses on these key features—specifically, blocks_accessed and avg_row_lengthA_var. This makes our heuristic comparable to those typically developed through domain observation, which often involves identifying significant features and determining their thresholds.

```python
def heuristic(input):
  if blocks_accessed <= 0.03894999995827675:
    if avg_row_lengthA_var <= 0.009800000116229057:
      predicted.append('2')
    elif avg_row_lengthA_var > 0.009800000116229057:
      predicted.append('1')
  elif blocks_accessed > 0.03894999995827675:
    if blocks_accessed <= 0.04165000095963478:
      predicted.append('0')
    elif blocks_accessed > 0.04165000095963478:
      predicted.append('0')
```

Listing 1: Decision-tree-guided heuristic function derived from decision trees

## VI. EVALUATION

*Decision Tree*– Figure 5 presents a comparison of decision tree performance against IP, OP, and RW methods, along with the heuristic approach. Although our test set comprises

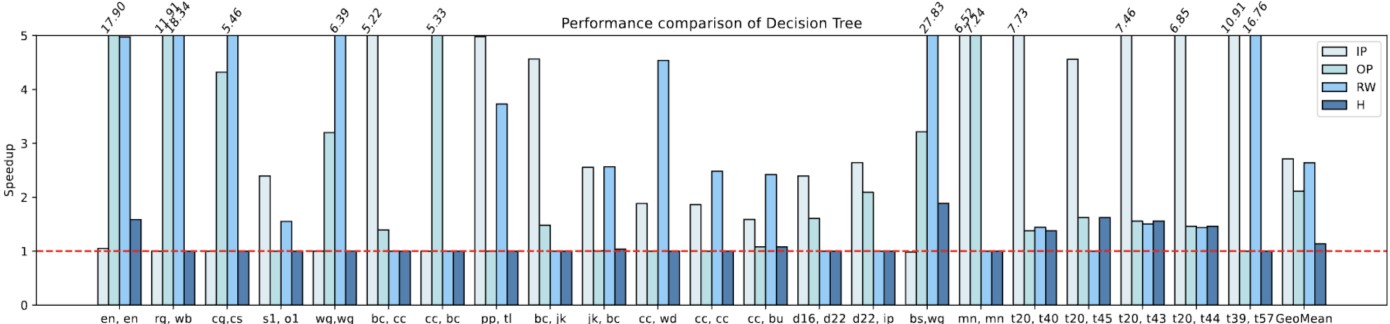

Fig. 5: **Performance of Decision Tree Model –** The speedup of the decision tree model over applying IP, OP, RW, and heuristic (H) for SpGEMM operations. Note: The y-axis has been truncated for both the graphs, and the bars are labeled with numbers on top to indicate the actual speedup values.

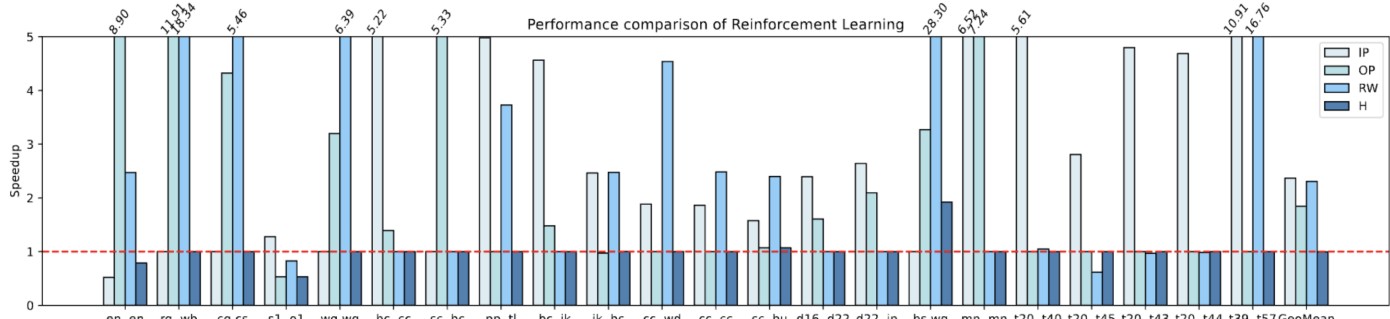

Fig. 6: **Performance of Reinforcement Learning Model –** The speedup of the reinforcement tree model over applying IP, OP, RW, and heuristic (H) for SpGEMM operations.

numerous SpGEMM data points, we have plotted only a select few due to space constraints. In Figure 5, the red line highlights a speedup of 1 signifying instances where the decision tree performance matched that of an existing dataflow scheme or the heuristic. We observed that most matrices from the SuiteSparse collection can be optimally processed using a single algorithm across all blocks. In such scenarios, the decision tree correctly identifies the consistent pattern and predicts the same optimal algorithm for all sub-matrices, resulting in a speedup of 1 for that specific dataflow scheme. Conversely, the generated random test matrices contain sub-matrices with varying patterns, leading to instances where different schemes excel for different blocks, as seen in data points (t20,t40), (t20,t43), and (t20,t44). Our k-fold cross-validation experiments yielded a 94% accuracy across the dataset, translating to average speedups of $2.7\times$ over the IP, $2.1\times$ over the OP, $2.64\times$ over the RW, and $1.13\times$ over the heuristic approach.

*Reinforcement Learning–* Figure 6 shows the performance of the RL model compared to the decision tree approach. Certain datapoints display a speedup of less than 1, suggesting that the RL model occasionally selected suboptimal dataflow schemes for specific matrix block during the SpGEMM computation. The accuracy analysis of the RL model across all test data achieved 90%, which is lower than that of the decision tree, explaining the observed discrepancies in speedups. On average, the RL model achieved a speedup of $2.36\times$ over IP,

$1.85\times$ over OP, and $2.3\times$ over RW. The heuristic's performance was comparable to that of the RL model.

*Storage–* Storage considerations are crucial when developing ML systems to ensure resource efficiency, low energy consumption, and faster inference latency. To address these concerns, we analyzed the storage requirements of our models and the heuristic file. The decision tree model, when simplified into a series of if-else statements—a process similar to our heuristic derivation—reduces its storage footprint from 29KB to 24KB by eliminating unnecessary metadata. This model is effectively converted into a code function that processes matrix feature inputs directly. In contrast, the RL model occupies 38KB and cannot be further reduced due to essential metadata about the RL environment and reward function. The heuristic is the most storage-efficient, requiring only 512B.

## VII. CONCLUSION

*ML or Heuristics?* While heuristics may seem like an appealing option over ML due to comparable performance and significantly lower storage requirements, the suitability of this approach depends on the specific needs of the system. If the system requires a very lightweight approach and is not expected to undergo significant changes—such as the addition or removal of components—then heuristics are an ideal choice. However, if the system is likely to evolve, ML models, particularly RL with online learning capabilities, are better equipped to adapt to changes quickly. We can

further enhance this adaptability by providing feedback to the agent to consider additional environmental parameters, such as PE utilization and system bandwidth. Another point of consideration is the potential to use ML to develop heuristics for systems. Our heuristic was derived from a decision tree that helped us identify the most influential features and the necessary thresholds. We can also consider increasing storage footprint of our heuristic to boost its accuracy by including additional features, thereby aligning its performance more closely with that of the models. This trade-off between storage and accuracy merits careful consideration in the development of efficient and effective heuristics for dynamic systems.

*Future Direction–* Our research aimed to investigate the application of ML in selecting dataflows for SpGEMM. We explored three distinct approaches: decision trees, RL models, and heuristics. This study pioneered in its examination of techniques to identify the most optimal dataflow in SpGEMM. However, this is just the beginning in this area of study. Future enhancements could include adding more data points to our dataset to refine the accuracy of our models and using that to expand our analysis of feature relationships. Furthermore, we plan to explore online reinforcement learning techniques and prioritize state pruning to reduce the storage demands of our RL model. We also aspire to deploy our technique across systems that support a range of SpGEMM algorithms.

## ACKNOWLEDGEMENTS

We gratefully acknowledge the support of US Department of Energy (DoE) under the ASCR ECRP, Award DE-SC0024079.

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
