# OpenReview forum: "Misam: Using ML in Dataflow Selection of Sparse-Sparse Matrix Multiplication"
_iscaconf.org/ISCA/2024/Workshop/MLArchSys — MLArchSys 2024 OralPoster_

### Official Review · Reviewer_B4MH · 2024-05-27
**A novel use of ML techniques for optimizing dataflow selection in hardware accelerators but the paper needs a better experimental setup**

**Confidence:** 4
**Rating:** 5

**Detailed Feedback And Questions For Authors:**

**Summary:** The paper presents a novel approach to optimize the execution of sparse matrix-matrix multiplication (SpGEMM) by utilizing machine learning techniques. The authors propose using decision trees and deep reinforcement learning to dynamically select the most appropriate dataflow scheme for various sparsity patterns. This adaptive method aims to outperform traditional heuristic-based approaches.

**Strengths:**
* The paper presents a novel use of ML techniques for optimizing dataflow selection in hardware accelerators.
* The paper includes a detailed explanation of dataset creation, feature selection, and the training process for both decision trees and reinforcement learning models.

**Weaknesses:**
* The dataset used for the experiments is relatively small, which may not fully capture the diversity of real-world applications.
* While the ML models show performance gains, the storage requirements are significantly higher compared to heuristic methods, which could limit its applicability in resource-constrained environments.

**Questions and Suggestions for the Authors:**
* Did the authors try/consider other ML techniques, such as random forests, support vector machines or neural networks?
* How does your approach compare with the latest advancements in adaptive dataflow selection, such as Spada and Flexagon?
* The paper would benefit from a wider range of ML techniques in evaluation.
* Using a larger dataset with more samples and more features to improve the ML techniques' accuracy would make the paper's claims stronger.

Overall, despite the weaknesses in the evaluation section, the main idea of using ML to select optimal dataflow schemes for SpGEMM is novel and the community would benefit from the ideas introduced in the paper.

**Top Reasons To Accept The Paper:**

* The paper presents a novel use of ML techniques for optimizing dataflow selection in hardware accelerators.
* The paper includes a detailed explanation of dataset creation, feature selection, and the training process for both decision trees and reinforcement learning models.

**Top Reasons To Reject The Paper:**

* The dataset used for the experiments is relatively small, which may not fully capture the diversity of real-world applications.
* While the ML models show performance gains, the storage requirements are significantly higher compared to heuristic methods, which could limit its applicability in resource-constrained environments.

---

### Official Review · Reviewer_JDcY · 2024-05-27
**Could be a solid paper if more technical details and strong results were provided**

**Confidence:** 3
**Rating:** 5

**Detailed Feedback And Questions For Authors:**

This paper could be a stronger paper with more technical details and stronger results. Please refer to the following for details:

- The cycle-accurate simulator needs further clarification: (1) does it model the PE array only or an entire system? (2) How is it validated? (3) Can it model different PE sizes beyond four? And so on.
- The SuiteSparse library needs a reference.
- More details regarding the workload could be provided; examples include the size matrices (beyond just sparsity ratio)
- Concrete prior works (e.g., Spada and Flexagon mentioned in this paper) could be compared.
- The plots in figures 5 and 6 are confusing; it can be read as individual methods (IP, OP, RW, and H) achieves presented speedups compared to a certain baseline. It would be better to have explicit latency bars that include this work (decision tree and RL).
- The heuristic seems to be very effective. Although Section VI justified the RL-based approach at a certain extent, the claims could be supported by quantitative analysis (e.g, adding evaluation scenarios with different system configuration that shows the limitation of the heuristics).

**Top Reasons To Accept The Paper:**

- The frontend is well written

**Top Reasons To Reject The Paper:**

- Many technical details need clarification (the cycle-accurate simulator, workload, and so on)
- The reported results are weak compared a simple heuristic discussed in the paper

---

### Official Review · Reviewer_JPq3 · 2024-05-28
**Misam: Analysis of using ML of algorithm selection for SpGEMM execution**

**Confidence:** 4
**Rating:** 6

**Detailed Feedback And Questions For Authors:**

Thanks for submitting the paper. The paper is well written and was easy to understand.
One minor comment:
1. Maybe I missed it - It was unclear what hardware was used for hardware performance measurement. Some discussion on how the feature selected would change depending on the hardware architecture would also have been interesting.

**Top Reasons To Accept The Paper:**

1. The paper targets an important problem of selecting key dataflow selection for SpGEMM.
2. The work does a good job of detailing the pros and cons of using ML for the selection policy and highlights the importance of adopting such techniques.
3. The work compares using RL and Decision trees for selection against different algorithms which delivers an average 2.6x gains

**Top Reasons To Reject The Paper:**

1. The work classifies any given SpGEMM into the three standard algorithms of IP,OP, and RW. It would have been good to select more advanced SpGEMM operations. That would also expand the feature space which would make the problem noteworthy.
2. Considering that the work is a competitor to single algorithms such as OuterSpace, it would have been good to show how much performance improvement it would have against such state-of-the-art SpGEMM techniques.
3. It wasn't clear what hardware was being used to evaluate the speedup.

---

### Official Review · Reviewer_EJKk · 2024-05-28
**The paper presents a machine learning-based approach for adaptively selecting the most appropriate dataflow scheme for SpGEMM tasks with diverse sparsity patterns.**

**Confidence:** 2
**Rating:** 4

**Detailed Feedback And Questions For Authors:**

Same as above comments in top reasons. Writing and details are unclear in the paper.

**Top Reasons To Accept The Paper:**

They have presented improvement in figure 6 using their dataflow selection approach; but I found missing information in figures and in papers to justify the results.

**Top Reasons To Reject The Paper:**

Some of the figures are very hard to interpret , for example figure 1 where size of the bubbles and color depth are used to read essential information which is hard to read.

Author did not mention what execution dataflow schemes are used for comparison in key figure 6 for IP, OP, and Row.

it is unclear on how do author come up with the threashold value fore decision trees, which will be an important factor.

Authors talked about reducing storage using decision tree but not much details on how to achieve this.

---

### Decision · Program_Chairs · 2024-05-30

**Decision:**

Accept (Oral/Poster)

**Comment:**

Congratulations! We are pleased to inform you that your paper has been accepted for presentation at MLArchSys 2024. We look forward to your participation at the workshop. Further details regarding the schedule and format will be provided soon. See you at the workshop!